# GUESS THE INSTRUCTION!
# FLIPPED LEARNING MAKES LANGUAGE MODELS STRONGER ZERO-SHOT LEARNERS

**Seonghyeon Ye**[1][*]    **Doyoung Kim**[1]    **Joel Jang**[1][*]    **Joongbo Shin**[2]    **Minjoon Seo**[1]

[1]KAIST    [2]LG AI Research
{seonghyeon.ye,ikevin98,joeljang,minjoon}@kaist.ac.kr
jb.shin@lgresearch.ai

## ABSTRACT

Meta-training, which fine-tunes the language model (LM) on various downstream tasks by maximizing the likelihood of the target label given the *task instruction* and input instance, has improved the zero-shot task generalization performance. However, meta-trained LMs still struggle to generalize to challenging tasks containing novel labels unseen during meta-training. In this paper, we propose FLIPPED LEARNING, an alternative method of meta-training which trains the LM to generate the task instruction given the input instance and label. During inference, the LM trained with FLIPPED LEARNING, referred to as FLIPPED, selects the label option that is most likely to generate the task instruction. On 14 tasks of the BIG-bench benchmark, the 11B-sized FLIPPED outperforms zero-shot T0-11B (Sanh et al., 2021) and even a 16 times larger 3-shot GPT-3 (175B) (Brown et al., 2020) on average by 8.4% and 9.7% points, respectively. FLIPPED gives particularly large improvements on tasks with unseen labels, outperforming T0-11B by up to +20% average F1 score. This indicates that the strong task generalization of FLIPPED comes from improved generalization to novel labels. We release our code at github.com/seonghyeonye/Flipped-Learning.

## 1 INTRODUCTION

Large Language Models (LMs) pretrained on a vast amount of corpora are capable of solving various downstream tasks through instructions (task prompts) concatenated with the input instances without any task-specific fine-tuning (Brown et al., 2020; Rae et al., 2021; Chowdhery et al., 2022; Zhang et al., 2022). Previous work has shown that fine-tuning the LM on various downstream tasks by generating the correct answer given a prompted input (instruction and input), also referred to as *meta-training*, leads to significant improvement in zero-shot task generalization (Sanh et al., 2021; Wei et al., 2021; Wang et al., 2022). However, Webson & Pavlick (2021); Min et al. (2022c) show that LMs meta-trained through this standard approach are sensitive to different label words, implying that standard meta-trained LMs often fail to generalize to tasks that contain novel labels.

In this paper, we introduce an alternative meta-training method called FLIPPED LEARNING that flips the task instruction and label space, training the underlying LM to generate the *instruction* when given the input instance and label. This differs from the standard meta-training methods which train the LM to generate the label given instruction and input instance (DIRECT) or generate instruction and input instance given the label (CHANNEL). Also, we add an unlikelihood loss for FLIPPED LEARNING, making the LM not generate the task instruction for an incorrect label option. During inference, the LM trained via FLIPPED LEARNING, referred to as FLIPPED, selects the label option that is most likely to generate the task instruction, as shown in Figure 1.

To compare with an existing meta-trained LM T0 (Sanh et al., 2021) trained by the DIRECT approach, we implement FLIPPED by meta-training the T5 (Raffel et al., 2019) model on 20 different

---

[*] Work done while interning at LG AI Research.

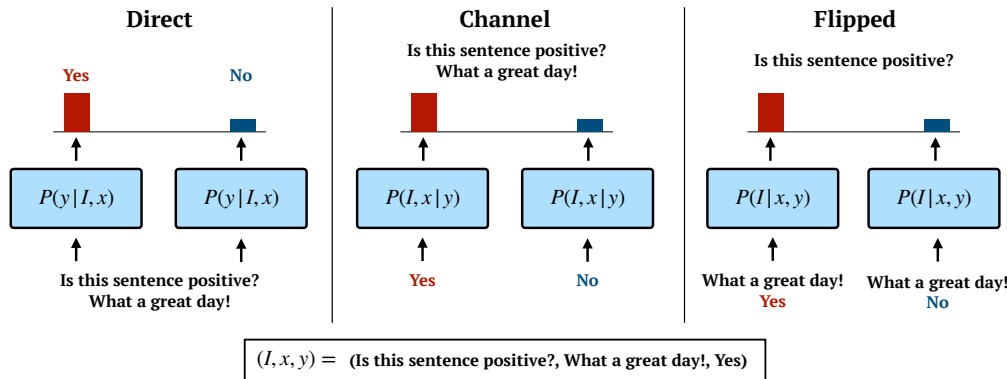

Figure 1: Inference of DIRECT, CHANNEL and FLIPPED to select an appropriate label (Yes) from label options (Yes/No). DIRECT, which is the standard LM inference, computes the conditional probability of label given instruction+input. CHANNEL, which is noisy channel inference, computes the conditional probability of instruction+input given label. Our FLIPPED computes the conditional probability of instruction given input+label.

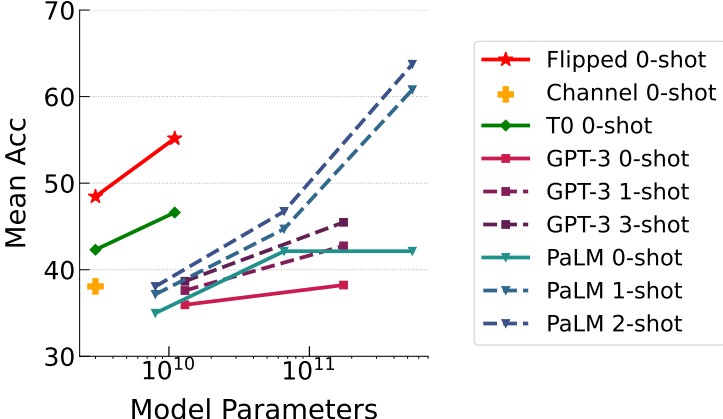

Figure 2: Mean Accuracy on 14 datasets from the BIG-Bench benchmark. FLIPPED shows the best performance among zero-shot LMs and even better performance than GPT-3 175B 3-shot.

datasets (around half of the datasets used to train T0) with only ∼5% of training compute compared to T0. Evaluation on 14 datasets from BIG-Bench (Srivastava et al., 2022) demonstrate that FLIPPED is effective (Figure 2), not only showing state-of-the-art performance compared to all LMs regardless of size in the zero-shot setting, but also outperforming much larger GPT-3 175B (3-shot) by a significant margin, even without any demonstrations of the task (zero-shot). We also compare FLIPPED with baseline models on 14 additional common English NLP tasks, further amplifying its effectiveness compared to previous methods and models.

We hypothesize that FLIPPED shows strong zero-shot generalization ability on unseen tasks because of the improved generalization capability to unseen *labels*. To test this hypothesis, we evaluate on various label pairs with different surface forms but with the same meaning (e.g. yes/no vs agree/disagree). Results show FLIPPED has up to +20% average F1 score performance gap with T0-11B, indicating that FLIPPED LEARNING indeed significantly improves label generalization capability. This hypothesis is further bolstered by the fact that the tasks that show significant performance improvement from the baselines among the 28 evaluation datasets are datasets with unseen labels during meta-training. Because FLIPPED LEARNING conditions on the label instead of generating it, FLIPPED LEARNING is likely to avoid label overfitting, resulting in improved label generalization, which consequently leads to better task generalization.

In summary, our contributions are as follows:

- We propose FLIPPED LEARNING, a novel meta-training method that computes the likelihood of the task instruction given the concatenation of input instance and label. By adding

an unlikelihood loss, we make LMs generate the task instruction depending on the input instance-label correspondence.

- On 14 datasets from the BIG-Bench benchmark, we show that 11B-sized FLIPPED (LM trained through FLIPPED LEARNING) outperforms not only meta-trained T0-11B by 8.4% points on average, but also 16x larger 3-shot GPT-3 by 9.7% points. When evaluating on 14 additional English NLP tasks, FLIPPED outperforms all baseline models on average, further demonstrating the effectiveness of our proposed method.

- We show that FLIPPED is particularly effective on generalization to labels that are unseen during meta-training, outperforming T0-11B by up to 20% average F1 score for novel label pairs.

## 2 RELATED WORK

### 2.1 META-TRAINING

Prior work has shown that *meta-training*, multitask fine-tuning on various downstream tasks with task instructions included, enables zero-shot task generalization (Sanh et al., 2021; Wei et al., 2021; Wang et al., 2022; Mishra et al., 2022). Specifically, Sanh et al. (2021); Wang et al. (2022) have shown that moderate-sized LMs can also generalize to unseen tasks through meta-training and the generalization performance improves by scaling the number of training tasks, the number of prompts per task, and the size of the LM. Based on this method, Ouyang et al. (2022) apply reinforcement learning with human feedback after meta-training to make better instruction-following LMs. To improve the task generalization performance of meta-trained LMs, Lin et al. (2022); Ye et al. (2022) suggest using a retrieval-based framework. Min et al. (2022b); Chan et al. (2022); Chen et al. (2021) apply meta-training by using input-label pairs instead of task instructions.

### 2.2 NOISY CHANNEL PROMPTING

When performing classification tasks, zero-shot LMs (Brown et al., 2020; Chowdhery et al., 2022) compute the conditional probability of the labels given input instances concatenated with instructions or demonstrations, referred to as *direct prompting*. On the other hand, *noisy channel prompting* reverts the input and the output space, making LMs generate every word in the input instances when conditioned on the label (Min et al., 2022a; Lazaridou et al., 2022). Specifically, Min et al. (2022b) apply noisy channel prompting during meta-training, optimizing the model to generate the input instance given the concatenation of demonstrations and the label. Motivated from Min et al. (2022b), we optimize the model to generate *only* the task instruction while conditioning on the input and label (example shown in Figure 1). While Honovich et al. (2022); Gupta et al. (2022) have similar intuition of guessing the instruction given input and label, they only do *flipping* on either training or inference, not both.

### 2.3 LABEL GENERALIZATION

Previous work has shown that LMs are very sensitive to different label surface forms, indicating poor robustness. Zhao et al. (2021) show that even 175B-sized GPT-3 suffers from high sensitivity and propose contextual calibration to solve this issue. Holtzman et al. (2021); Shi et al. (2022) define this problem as surface form competition and propose Domain Conditional Pointwise Mutual Information scoring or fuzzy verbalizers to mitigate this problem. For meta-training, Webson & Pavlick (2021) analyze the effect of various label surface forms for a meta-trained LM and find that meta-trained LMs are more sensitive to label surface forms than different wordings of the prompt, which suggests that the meta-trained LMs *overfit* to the label space provided during meta-training. This shows that meta-trained LMs cannot generalize to unseen label space, indicating poor *label generalization*.

## 3 FLIPPED LEARNING

In this section, we introduce FLIPPED LEARNING, which trains the LM to compute the conditional probability of the task instruction given input instance and label (Figure 1). We first introduce

notations and compare the difference between previous approaches (DIRECT and CHANNEL) and our proposed method during inference (Section 3.1). Next, we provide a detailed explanation of the training objective of FLIPPED LEARNING during meta-training and explain the intuition for including an unlikelihood training loss in addition to the original loss (Section 3.2).

## 3.1 INFERENCE OF PROBABILISTIC LMs

In this work, we focus on tasks with label options such as classification and multi-choice tasks for both meta-training and evaluation. For a given task $T = \{x, Y\}$ where $x$ is the input instance and $Y = \{y_1, ...y_k\}$ is label option set, we convert the data instance into a prompted version $\{[I, x], L\}$. From $\{[I, x], L\}$, $[I, x]$ denotes the prompted input instance including natural language instruction $I$ and $L = \{l_1, ..., l_k\}$ denotes the natural language label option set where $l_i = v_I(y_i)$ and $v_I$ is the verbalizer corresponding to $I$. The goal during inference is to select the correct $l_i$ from $L = \{l_1...l_k\}$ given $I$ and $x$.

**DIRECT** method computes the conditional probability of the label given task instruction and input instance. During inference, it selects the label that leads to the highest conditional probability:

$$\arg \max_{l_i} P(l_i|I, x) \tag{1}$$

This is the most common approach used for zero-shot inference of LMs (Brown et al., 2020; Chowdhery et al., 2022; Sanh et al., 2021; Wei et al., 2021).

**CHANNEL** method (Min et al., 2022a) computes the conditional probability of instruction and input instance given a label. Using Bayes' rule, the probability can be reparameterized as follows:

$$\arg \max_{l_i} P(l_i|I, x) = \arg \max_{l_i} \frac{P(I, x|l_i)P(l_i)}{P(I, x)} = \arg \max_{l_i} P(I, x|l_i) \tag{2}$$

since $P(I, x)$ is independent from $l_i$ and $P(l_i) = \frac{1}{|L|}$; we assume the prior to be an uniform distribution for tasks with label options.

**FLIPPED LEARNING**, our proposed method, computes the conditional probability of the task instruction given an input instance and a label. Different from previous approaches, we separate $[I, x]$ into $I$ and $x$ and use Bayes' rule to reparameterize the conditional probability as follows:

$$\arg \max_{l_i} P(l_i|I, x) = \arg \max_{l_i} \frac{P(I|x, l_i)P(l_i, x)}{P(I, x)} = \arg \max_{l_i} P(I|x, l_i)P(l_i|x) \approx \arg \max_{l_i} P(I|x, l_i) \tag{3}$$

where we assume $P(l_i|x) \approx \frac{1}{|L|}$ for simplicity. By considering $P(I|x, l_i)$, we allow the LM to put more focus on the task instruction. The intuition of FLIPPED LEARNING can be considered to be similar to generative question answering (Lewis & Fan, 2018) which generates the question given context and answer, but FLIPPED LEARNING generates the task instruction for task generalization.

To compute $P(I|x, l_i)$ for FLIPPED LEARNING, the prompted input $[I, x]$ should be separated into task instruction $I$ and input instance $x$. However, the prompted input might be sometimes an intermix of $I$ and $x$ rather than their sequential concatenation. To handle this, we follow Raffel et al. (2019)'s denoising objective using sentinel tokens where the portions representing task instruction $I$ are replaced by sentinel tokens and the LM learns to denoise the sentinel tokens by generating $I$.[1]

## 3.2 META-TRAINING USING FLIPPED LEARNING

Next, we explain how we optimize the LM to utilize $P(I|x, l_i)$ which requires adding in an *unlikelihood* loss during meta-training. Given the sequence of task instruction $I = (I_1, .., I_T)$, we denote the LM loss function as follows:

$$L_{LM} = -\sum_{t=1}^{T} \log P(I_t|x, l_c, I_{<t}) \tag{4}$$

where $l_c$ corresponds to the correct label option. By minimizing this loss function, the LM learns to generate $I$ when given the correct label option and the input instance.

---

[1] We illustrate the denoising objective example in Appendix C.

**Unlikelihood Loss**  However, from preliminary experiments, we observe that meta-training the LM only on $L_{LM}$ results in ignoring the correspondence between the *input instance* and *label*: meta-trained LM generates task instruction $I$ regardless of the correspondence of the label option. We conjecture that this is a result of shortcut learning of large LMs (Du et al., 2022; Min et al., 2022c). To amplify the correspondence signal between the input instance and the correct label, we add an unlikelihood loss (Tam et al., 2021; Liu et al., 2022; Welleck et al., 2019) during meta-training which can be denoted as follows:

$$L_{UL} = -\sum_{t=1}^{T} \log(1 - P(I_t | x, l_{c'}, I_{<t})) \tag{5}$$

where $l_{c'}$ corresponds to an incorrect label option randomly sampled from the incorrect label option set $L_{C'} = \{l | l \in L, l \neq l_c\}$. This unlikelihood loss term allows the LM to *not* generate the task instruction if the label option does not correspond to the input instances. The final training objective of FLIPPED LEARNING is the weighted sum of $L_{LM}$ and $L_{UL}$:

$$L = L_{LM} + \lambda L_{UL} \tag{6}$$

where $\lambda$ is a hyperparameter. By optimizing both likelihood and unlikelihood objectives, the LM is optimized to generate the instruction when given the correct label and not generate the instruction when given the incorrect label, strengthening the correspondence between the input instance and the correct label.

## 4  EXPERIMENTAL SETUP

**Training**  For meta-training, we utilize the subset of T0 (Sanh et al., 2021) meta-training datasets: 4 task clusters (sentiment classification, paraphrase detection, topic classification, multi-choice QA), which are 20 datasets in total. We only train on tasks with label options and exclude tasks such as free-form generation because FLIPPED LEARNING requires label options for unlikelihood training on incorrect label options. We provide detailed training configurations in Appendix D and the full list of training datasets in Appendix E.1.

**Evaluation**  Following the evaluation setting of Sanh et al. (2021), we first measure unseen task generalization performance on 14 tasks of BIG-bench which contain challenging and various tasks that are unseen during meta-training. For each of the BIG-bench tasks, we report the accuracy of a single instruction for each task following the convention of past work (Sanh et al., 2021; Lin et al., 2022). Furthermore, we additionally evaluate on 14 English NLP unseen tasks, consisting of 7 classification and 7 multi-choice datasets, also following the setting of Sanh et al. (2021); Lin et al. (2022). For evaluation metric, we use Macro-F1[2] for classification and accuracy for multi-choice tasks, following Min et al. (2022b;c). We also report the average standard deviation among different evaluation instructions, indicating the robustness of different wordings of the evaluation instruction (the lower, the better). For analysis of label generalization of classification tasks, we evaluate on 5 datasets: 2 seen datasets during meta-training (IMDB, PAWS) and 3 unseen datasets (RTE, CB, WiC). We provide the full list of evaluation datasets in Appendix E.2 and more details on the evaluation setting are specified in Appendix F.

**Baselines**  We evaluate several baselines to observe the effectiveness of FLIPPED LEARNING: (1) T0-3B, a 3B-sized meta-trained LM by Sanh et al. (2021), (2) DIRECT, a 3B-sized meta-trained LM using the same language modeling objective (standard method) of T0-3B, but with our training configurations, (3) CHANNEL, a 3B-sized meta-trained LM using noisy channel language modeling objective, (4) FLIPPED-3B, a 3B-sized LM meta-trained through our proposed FLIPPED LEARNING, (5) T0-11B, a larger meta-trained LM of Sanh et al. (2021), (6) FLIPPED-11B, a larger meta-trained LM trained through FLIPPED LEARNING, (7) GPT-3 (Brown et al., 2020), 175B sized pretrained LM, (8) PaLM (Chowdhery et al., 2022), 540B sized pretrained LM. Note that DIRECT, CHANNEL, and FLIPPED is meta-trained on the same number of datasets and training steps.

---

[2]Macro-F1 is more appropriate for imbalanced classification than accuracy.

| Dataset (metric) | Zero-shot | | | | | | | | Few-shot | |
|---|---|---|---|---|---|---|---|---|---|---|
| | T0 3B | DIR. 3B | CHAN. 3B | FLIP. 3B | T0 11B | FLIP. 11B | GPT-3 175B | PALM 540B | GPT-3 (3) 175B | PALM (1) 540B |
| Known Un. | 47.83 | 63.04 | 52.17 | 71.74 | 58.70 | **86.96** | 60.87 | 56.52 | 50.00 | 67.39 |
| Logic Grid | 41.10 | 35.90 | 30.90 | 41.70 | 38.30 | **42.50** | 31.20 | 32.10 | 31.10 | 42.20 |
| Strategy. | 52.79 | 53.28 | 53.01 | 53.19 | 52.75 | 53.23 | 52.30 | **64.00** | 57.10 | 69.00 |
| Hindu Kn. | 25.71 | 50.29 | 16.57 | 47.43 | 29.71 | 52.57 | 32.57 | **56.00** | 58.29 | 94.86 |
| Movie D. | 52.85 | 47.15 | 51.06 | 47.93 | **53.69** | 48.49 | 51.40 | 49.10 | 49.40 | 57.20 |
| Code D. | 46.67 | 33.33 | **71.67** | 45.00 | 43.33 | 60.00 | 31.67 | 25.00 | 31.67 | 61.67 |
| Concept | 45.52 | 58.14 | 35.67 | 61.64 | **69.29** | 64.93 | 26.78 | 59.26 | 35.75 | 80.02 |
| Language | 14.84 | 22.01 | 11.55 | 19.01 | 20.20 | **26.87** | 15.90 | 20.10 | 10.90 | 37.30 |
| Vitamin | 58.89 | 63.83 | 15.73 | 57.07 | 64.73 | **65.57** | 12.30 | 14.10 | 52.70 | 70.40 |
| Syllogism | **52.94** | 49.85 | 50.43 | 50.56 | 51.81 | 50.39 | 50.50 | 49.90 | 52.80 | 52.20 |
| Misconcept. | 50.23 | 50.23 | 47.79 | 46.58 | 50.00 | **54.34** | 47.95 | 47.49 | 60.27 | 77.63 |
| Logical | 46.64 | 38.06 | 25.73 | 59.82 | 54.86 | **64.56** | 23.42 | 24.22 | 33.93 | 34.42 |
| Winowhy | 44.29 | 44.33 | **55.36** | 53.33 | 52.11 | 55.08 | 51.50 | 45.30 | 56.50 | 47.50 |
| Novel Con. | 15.63 | 3.13 | 15.63 | 25.00 | 15.63 | **46.88** | **46.88** | **46.88** | 56.25 | 59.38 |
| BIG-bench AVG | 42.56 | 43.75 | 38.07 | 48.57 | 46.79 | **55.17** | 38.23 | 42.14 | 45.48 | 60.80 |

Table 1: Task generalization performance on 14 BIG-bench tasks. DIR. denotes DIRECT, CHAN. denotes CHANNEL, and FLIP. denotes FLIPPED. Parentheses in the *Few-shot* column denote the number of shots. FLIPPED performs the best on average for zero-shot setting.

## 5 EXPERIMENTAL RESULTS

In this section, we evaluate the effectiveness of FLIPPED compared to various baselines. For task generalization performance, we evaluate on 14 tasks of BIG-bench and 14 common English NLP tasks (Section 5.1). For analysis of FLIPPED, we evaluate on multiple label options with the same meaning but with different surface forms (Section 5.2).

### 5.1 MAIN RESULTS

**DIRECT outperforms T0-3B.** Our implementation of DIRECT outperforms T0-3B significantly despite using only half of the datasets used to train T0 (20 out of 38), indicating competitive performance of our baselines. Note that T0-3B and DIRECT are trained on the same training objective, with the only difference in training configurations, showing that our training setting is more optimal to evaluate the task generalization performance of meta-trained LMs. We conjecture the significant performance improvement to come from 3 potential factors: (1) We train for about 5% token updates compared to Sanh et al. (2021), implying that DIRECT avoids overfitting to the training dataset (2) DIRECT includes the EOS (end-of-sequence) token during training and evaluation unlike T0-3B, (3) We do not use sequence-packing during meta-training. Our DIRECT baseline shows that the task generalization performance from Sanh et al. (2021) might have been underestimated [3].

**CHANNEL is not effective for task generalization.** CHANNEL method largely underperforms DIRECT and T0, showing close to random guessing performance for many unseen tasks. This result is consistent with that of Min et al. (2022b) in that instructions on zero-shot (not few-shot) setting *worsen* the task generalization performance for CHANNEL method unlike the DIRECT method. We conjecture that because prompted inputs are mostly question-answer formats, it is unnatural for CHANNEL to generate a question-like instruction given only an answer even after meta-training.

**FLIPPED outperforms baselines.** For the 14 BIG-bench tasks of Table 1, FLIPPED-3B significantly outperforms all meta-trained models with the same model size: +6.01% mean accuracy compared to T0-3B and +4.82% mean accuracy compared to DIRECT. FLIPPED-3B also outperforms 4x

---

[3]Although one might think that the other 18 training datasets of T0 might have had a negative impact on unseen tasks, we observe that training on whole datasets also shows similar results to DIRECT from preliminary experiments, implying that the performance improvement comes from the 3 potential factors mentioned above.

| Dataset (metric) | T0 3B | Dir. 3B | Chan. 3B | Flip. 3B | T0 11B | Flip. 11B | GPT-3 175B |
|---|---|---|---|---|---|---|---|
| RTE (F1) | 61.89 | 72.83 | 36.62 | 71.03 | **80.91** | 72.20 | 40.68 |
| CB (F1) | 30.94 | 49.81 | 22.35 | 52.27 | 53.82 | **61.51** | 29.72 |
| ANLI R1 (F1) | 24.39 | 30.17 | 21.30 | 33.92 | 34.72 | **34.93** | 20.90 |
| ANLI R2 (F1) | 23.73 | 28.23 | 21.44 | **32.62** | 31.25 | 32.59 | 22.50 |
| ANLI R3 (F1) | 23.45 | 30.41 | 22.50 | 34.65 | 33.84 | **34.77** | 23.77 |
| WSC (F1) | 54.64 | 50.35 | 46.38 | 52.82 | **58.36** | 49.88 | 26.24 |
| WiC (F1) | 38.53 | 36.42 | 38.69 | 37.36 | **51.64** | 39.26 | 45.36 |
| COPA | 75.88 | 89.63 | 50.13 | 89.88 | **91.50** | 90.75 | 91.00 |
| Hellaswag | 27.43 | 31.61 | 20.82 | 41.64 | 33.05 | 41.97 | **78.90** |
| StoryCloze | 84.03 | 94.24 | 57.84 | 95.88 | 92.40 | **96.12** | 83.20 |
| Winogrande | 50.97 | 55.96 | 50.99 | 58.56 | 59.94 | 66.57 | **70.20** |
| PIQA | 56.63 | 62.60 | 47.08 | 67.32 | 67.67 | 71.65 | **81.00** |
| ARC-Chall | 51.10 | 49.30 | 29.23 | 49.63 | 56.99 | **64.62** | 51.40 |
| OpenbookQA | 42.66 | 54.00 | 38.57 | 62.11 | 59.11 | **72.54** | 68.80 |
| En NLP AVG | 46.16 | 52.54 | 36.00 | 55.69 | 57.51 | **59.24** | 52.41 |
| En NLP STD (↓) | 4.74 | 4.36 | 4.58 | 3.29 | 5.24 | **3.11** | - |

Table 2: Zero-shot task generalization performance on 14 English NLP tasks consisted of 7 classification and 7 multi-choice tasks. 11B-sized FLIPPED (FLIP.) shows the best performance on average and also shows the best robustness to different evaluation instructions (lower STD).

times larger meta-trained T0-11B on average by +1.78% points. This result is significant considering that the effect of scaling law is strong for zero-shot generalization of meta-trained models (Wei et al., 2021; Sanh et al., 2021; Wei et al., 2022). FLIPPED-11B even shows better performance, outperforming T0-11B on average by +8.38% points. Compared to even larger pretrained LMs evaluated in a few-shot setting, FLIPPED-11B outperforms 3-shot GPT-3 which is 16x larger by 9.69% points on average. When compared to 1-shot PaLM which is 50x larger, FLIPPED outperforms on 4 tasks out of the 14 tasks. This shows that FLIPPED is effective for generalizing to unseen tasks that are challenging, resulting in the best performance on the zero-shot setting even when compared to LMs with much larger sizes.

For the 14 common English NLP tasks which are consisted of 7 classification and 7 multi-choice tasks shown in Table 2, FLIPPED-3B outperforms baseline models with the same model size (T0-3B, DIRECT, CHANNEL) on task generalization performance by a significant margin, largely reducing the gap between T0-11B. FLIPPED-11B shows the best performance on average, outperforming T0-11B by 1.73% points. Also, FLIPPED shows the lowest standard deviation among multiple different evaluation instructions compared to other meta-trained baseline models, including T0-11B. This indicates that FLIPPED is not only effective for zero-shot task generalization but also *robust* to different surface forms of the instruction.

## 5.2 ANALYSIS OF FLIPPED

**FLIPPED significantly outperforms baselines for tasks with unseen labels.** We first analyze the tasks that FLIPPED outperforms baseline models and identify a clear correlation: FLIPPED especially shows strong performance on tasks that contain many label options unseen during meta-training. For RTE, WSC, WiC datasets, which are consisted of *seen* label options (yes/no), direct meta-trained LMs (DIRECT, T0) show strong performance as shown in Table 2. On the other hand, FLIPPED shows strong performance on CB and ANLI datasets that contain an *unseen* label (e.g. maybe). This can be seen as a result of effective label generalization of FLIPPED; the prediction of T0 and DIRECT is largely biased to label options that are seen during training (yes/no) while FLIPPED makes balanced predictions (yes/no/maybe). Although the calibration method of Zhao et al. (2021) is known to mitigate the prediction bias of LMs, we find that calibration worsens the performance of meta-trained LMs (Appendix G).

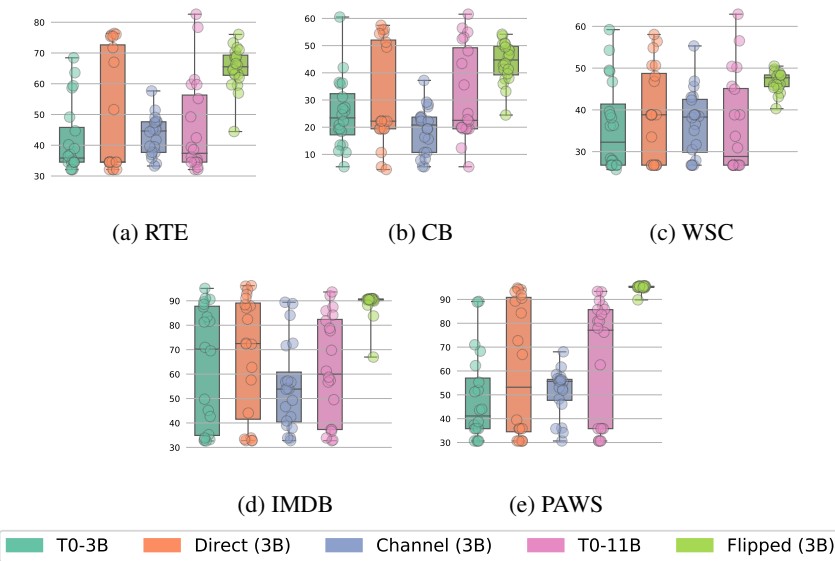

(a) RTE        (b) CB        (c) WSC

(d) IMDB        (e) PAWS

| T0-3B | Direct (3B) | Channel (3B) | T0-11B | Flipped (3B) |

Figure 3: Label generalization performance on 3 unseen and 2 seen datasets during meta-training. We evaluate on 20 different label pairs including many unseen labels. Result shows that FLIPPED significantly outperforms other baseline models.

For multi-choice tasks in Table 2, which are more likely to contain many unseen labels because they have different label options for every data instance, FLIPPED outperforms meta-trained LMs with the same size for most tasks. Moreover, FLIPPED outperforms T0-11B on BIG-bench (Table 1) which is mostly consisted of unseen labels. From these findings, we can hypothesize that the strong zero-shot task generalization of FLIPPED is likely to come from its strong label generalization capability.

**FLIPPED generalizes to unseen labels that are semantically the same.** To further test the above hypothesis, we analyze the label generalization performance of FLIPPED compared to other baseline models by varying the surface form the label options (e.g. yes/no vs agree/disagree) for 5 classification datasets: 3 datasets (RTE, CB, WSC) for unseen tasks, and 2 datasets (IMDB, PAWS) for seen tasks during meta-training. We vary the label options to 20 different pairs that have the same meaning but different surface forms including the original labels.[4]

Figure 3 shows the label generalization performance of T0-3B, DIRECT, T0-11B and FLIPPED-3B. For unseen tasks, FLIPPED outperforms T0-3B by (+23.37%, +18.78%, +10.92%), outperforms DIRECT by (+16.42%, +13.46%, +7.82%), and outperforms CHANNEL by (+21.88%, 24.84%, 9.93%) average F1 score on (RTE, CB, WSC) respectively. Even when compared with a 4x times larger meta-trained LM (T0-11B), FLIPPED outperforms by (+19.72%, +12.32%, +10.81%) average F1 score for (RTE, CB, WSC) respectively. This shows that FLIPPED can generalize to various novel labels, which is what even larger meta-trained LMs trained through direct prompting cannot do. Although baseline models outperform FLIPPED for best accuracy among different label pairs, this is mostly when the label is seen during meta-training (e.g. yes/no). The result of Figure 3 also indicates that the classification tasks evaluation setting of Sanh et al. (2021) overestimates the true generalization ability of LMs because Sanh et al. (2021) mostly evaluate unseen target tasks on labels that are *seen* during meta-training (yes/no), which is not guaranteed for a *true* zero-shot generalization scenario.

Aligned with the experiments on the 3 unseen tasks, FLIPPED further outperforms baselines on the 2 *seen* tasks during meta-training by a significant margin: (+25.55%, +46.68%) for T0-3B, (+20.74%, +34.66%) for DIRECT, (+34.12%, +43.74%) for CHANNEL, and (+26.63%, +31.91%) for T0-11B on (IMDB, PAWS). This further bolsters the hypothesis that standard meta-training leads to label overfitting, especially for seen tasks and FLIPPED LEARNING avoids this by conditioning on the label option instead of generating it.

---

[4]We collected synonyms from `https://www.thesaurus.com`. We provide the full list of 20 label options in Appendix H.

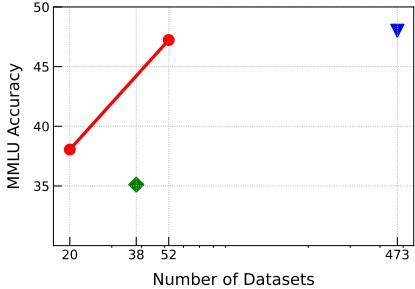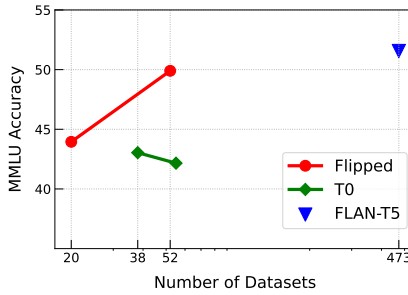

Figure 4: Zero-shot MMLU accuracy when scaling the number of datasets during meta-training. FLAN-T5 (Chung et al., 2022) is trained on 473 datasets (1,836 tasks). Although FLIPPED trained with 52 datasets (FLIPPED+) uses about 10% of training datasets compared to FLAN-T5, it largely reduces the performance gap. **Left**: Average accuracy of 3B-sized models on MMLU benchmark. **Right**: Average accuracy of 11B-sized models on MMLU benchmark.

## 6 ADDITIONAL EXPERIMENTS

Concurrent work of Chung et al. (2022) show that scaling the number of training datasets (up to 473 datasets) during meta-training results in state-of-the-art performance on challenging tasks such as the MMLU benchmark (Hendrycks et al., 2020). From the findings of Appendix A.2, we also expect that scaling up the number of datasets during meta-training can improve the performance further. Similar to the approach of Chung et al. (2022), we scale up the number of datasets during meta-training by adding generation tasks that are used to train the T0++ model (Sanh et al., 2021). For generation tasks, we train with the same training objective as classification tasks. For unlikelihood training of generation tasks, we sample an incorrect label option from a different training instance of the same dataset which is different from the correct label option. The number of training datasets in total is 52 and we refer to the model trained with FLIPPED LEARNING with these datasets as FLIPPED+. We evaluate FLIPPED+ on the zero-shot setting of the MMLU benchmark and compare the performance with T0 models (Sanh et al., 2021) and FLAN-T5 (Chung et al., 2022) on the same model size, shown in Figure 4.

Consistent with the result of Appendix A.2, FLIPPED LEARNING additionally benefits from the scale of the number of datasets: FLIPPED+ outperforms FLIPPED for both 3B and 11B sized models. Compared to T0 models, which do not always benefit from scaling the number of datasets, FLIPPED+ shows significant improvement. Moreover, while only using about 10% of the number of training datasets compared to FLAN-T5, FLIPPED+ largely reduces the performance gap between FLAN-T5. We suggest that using less number of training datasets during meta-training but resulting in strong zero-shot performance is important because it is closer to a *true* zero-shot setting.

## 7 CONCLUSION

In this paper, we propose FLIPPED LEARNING, which is a meta-training method that flips the instruction and label space, training the LM to compute the conditional probability of the task instruction given input instance and label. Our findings show that by conditioning on the label space instead of generating it, FLIPPED LEARNING avoids label overfitting, leading to better zero-shot unseen task generalization capabilities especially for tasks that contain various novel labels. To this end, we suggest the research community consider FLIPPED LEARNING for making efficient LMs that can generalize to challenging unseen tasks.

ACKNOWLEDGMENTS

We thank Sewon Min, Sungdong Kim, Miyoung Ko, Seungone Kim, Yujin Kim, and Yejin Cho for helpful discussions. This work was partly supported by Institute of Information & communications Technology Planning & Evaluation (IITP) grant funded by the Korea government (MSIT) (No.2022-0-00113, Developing a Sustainable Collaborative Multi-modal Lifelong Learning Framework, 80%; No.2021-0-02068, Artificial Intelligence Innovation Hub, 10%; No.2019-0-00075, Artificial Intelligence Graduate School Program (KAIST), 10%).

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

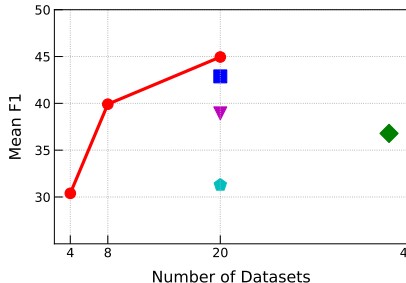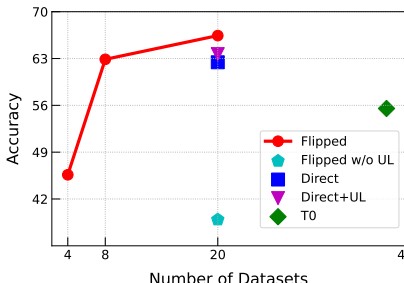

Figure 5: FLIPPED trained on varying numbers of datasets. FLIPPED W/O UL indicates ablation of FLIPPED without unlikelihood training. We also analyze the effect of unlikelihood training on DIRECT (DIRECT+UL). **Left**: Average F1 score of 7 classification tasks. **Right**: Average accuracy of 7 multi-choice tasks. All models are 3B-sized meta-trained LMs.

## A  ABLATION STUDIES

In this section, we analyze the effect of unlikelihood training. Also, we vary the number of meta-training datasets of FLIPPED to analyze the effect of the number of datasets on task generalization. We evaluate on 14 English NLP tasks and report average F1 score on 7 classification tasks and mean accuracy on 7 multi-choices tasks respectively.

### A.1  EFFECT OF UNLIKELIHOOD TRAINING

As mentioned in Section 3.2 and shown in Figure 5, we observe that FLIPPED LEARNING ignores the input instance-label correspondence if unlikelihood loss is not added, hurting the performance significantly. We additionally analyze if the strong task generalization of FLIPPED is solely coming from unlikelihood training by applying unlikelihood training on DIRECT, which is our strong baseline. As shown in the performance of DIRECT+UL in Figure 5, unlikelihood training worsens the task generalization performance especially for classification tasks while giving marginal improvement on multi-choice tasks, underperforming FLIPPED for both types of tasks. This shows that the effectiveness of FLIPPED LEARNING is not coming from unlikelihood training itself; both factors of FLIPPED LEARNING, flipping the label and instruction space and unlikelihood training, are needed to generalize effectively on unseen target tasks.

### A.2  NUMBER OF DATASETS

Meta-trained LMs via direct prompting shows improved performance when the number of datasets increases (Sanh et al., 2021; Wang et al., 2022; Wei et al., 2021). We also analyze if this phenomenon holds for FLIPPED LEARNING by varying the number of datasets per task cluster; we increase the total number of datasets by 4, 8, and 20. As shown in Figure 5, the performance of FLIPPED increases as the number of datasets increases, similar to LMs trained through direct prompting. Interestingly, using only 8 datasets to meta-train FLIPPED also shows strong performance, outperforming DIRECT model trained with 20 datasets on multi-choice tasks. Also, this efficient but effective model significantly outperforms T0-3B, while only using 20% of the number of datasets and 5% token updates. This shows that FLIPPED LEARNING can result in generalization to unseen tasks while using only a few number of datasets, making not only effective but also *efficient* zero-shot learners.

## B  LIMITATIONS

In this work, we do not explore FLIPPED for performing unseen tasks that do not have label options such as free-form generation. However, we believe FLIPPED can be used for these tasks as well by obtaining the list of label options from a different LM, which we leave for future work. FLIPPED LEARNING also assumes that the task instruction and input instance can be separated during zero-shot inference. However, although instruction-based benchmarks such as Natural Instructions (Mishra et al., 2022; Wang et al., 2022) define the prompted input as a naïve concatenation of

**<extra_id_0> Using only the above description and what you know about the world, is "<extra_id_1> " definitely correct? Yes or no?**

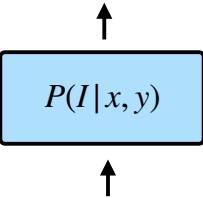

$P(I \,|\, x, y)$

**input: The girl was found in Drummondville.<extra_id_0>**
**Drummondville contains the girl.<extra_id_1>**
**output: Yes**

Figure 6: Illustration of denoising objective of FLIPPED LEARNING. Given, an input instance with sentinel tokens, FLIPPED LEARNING makes the LM generate the task instruction corresponding to the sentinel tokens for a correct label option.

task instruction and input instance, this is not guaranteed for prompt libraries such as Promptsource (Bach et al., 2022). Therefore, FLIPPED LEARNING needs additional techniques to separate the task instruction and the input instances as shown in Section 3.1.

## C  ILLUSTRATION OF DENOISING OBJECTIVE

As shown in Figure 6, FLIPPED LEARNING uses a denoising objective while meta-training to effectively separate the prompted input obtained through Promptsource (Bach et al., 2022) into task instruction and the input instance. By replacing task instruction as sentinel tokens, FLIPPED LEARNING makes the LM generate the task description that corresponds to the sentinel tokens.

## D  TRAINING CONFIGURATIONS

For backbone LM of FLIPPED, we use T5.1.1 (Raffel et al., 2019) which is pre-trained on a *denoising* objective while we use T5-LM adapted model (Lester et al., 2021) for DIRECT and CHANNEL which is continually trained T5.1.1 model on *language modeling* objective for 100B additional tokens. We use a different backbone LM for FLIPPED LEARNING because the meta-training objective is denoising objective while DIRECT and CHANNEL is language modeling objective. From preliminary experiments, we observe that the language modeling training objective of DIRECT and CHANNEL on T5.1.1 model leads to poor performance. Also, denoising objective of FLIPPED LEARNING on T5-LM adapted model leads to poor performance. Following Sanh et al. (2021); Raffel et al. (2019), we limit the number of data instances for each dataset to 500,000 to resolve data instance imbalance during meta-training. We train each model for 5K steps, with a batch size of 240. We set input and output sequence lengths as 512 and 128 respectively for FLIPPED-3B. For FLIPPED-11B, we set input and output sequence lengths as 384 and 64 respectively for computational efficiency. For DIRECT and CHANNEL, we set the learning rate as 1e-4 and for FLIPPED, we set the learning rate as 5e-5 because the training objective is different (generation vs denoising). We set the weight hyperparameter of likelihood and unlikelihood loss as $\lambda = 3$. Note that our total training compute used during meta-training is around 5% that of the training compute used to train the original T0: different from Sanh et al. (2021) which uses the batch size of 1024, sequence length of 1024, training steps of 12,200, we use a batch size of 240, half of the sequence length, training steps of 5,000 leading to 4.8% token updates compared to T0. For FLIPPED+, we almost keep the training configurations of FLIPPED with only a few variations. Unlike FLIPPED, we limit the number of data instances for each dataset to 50,000 to resolve data instance imbalance during meta-training. Also, for 3B-sized FLIPPED+, we train for 10K steps during meta-training due to the increased number of datasets. For 11B-size FLIPPED+, we keep the number of training steps to 5K steps due to computational costs.

|                      | Classification | Multi-choice |
| -------------------- | :------------: | :----------: |
| T0-3B                |     36.79      |    55.53     |
| T0-3B + Calibration  |     33.59      |    46.40     |
| FLIPPED              |   **44.95**    |  **66.43**   |

Table 3: Effect of calibration on T0-3B meta-trained LM. Results show that the performance worsens if calibration is applied especially for multi-choice tasks.

## E  TRAINING AND EVALUATION DATASETS

### E.1  META-TRAINING DATASETS

We use 4 task clusters for meta-training of DIRECT, CHANNEL and FLIPPED: sentiment classification, paraphrase, topic classification, which is 20 datasets in total. We use imdb (Maas et al., 2011), amazon_polarity (McAuley & Leskovec, 2013), rotten_tomatoes (Pang & Lee, 2005), yelp_review_full (Zhang et al., 2015b), app_reviews for sentiment, glue/qqp (Wang et al., 2018), paws/labeled_final (Zhang et al., 2019), glue/mrpc (Dolan & Brockett, 2005) for paraphrase, ag_news (Zhang et al., 2015a), dbpedia_14 (Lehmann et al., 2015) for topic classification, cos_e/v1.11 (Rajani et al., 2019), dream (Sun et al., 2019), quail (Rogers et al., 2020), quartz (Tafjord et al., 2019b), social_i_qa (Sap et al., 2019), wiqa (Tandon et al., 2019), cosmos_qa (Huang et al., 2019), qasc (Khot et al., 2020), quarel (Tafjord et al., 2019a), sciq (Welbl et al., 2017) for multi-choice QA.

### E.2  EVALUATION DATASETS

We evaluate on 14 datasets of BIG-bench benchmark (Srivastava et al., 2022): Known Unknown, Logic Grid, StrategyQA, Hindu Knowledge, Movie Dialog, Code Description, Conceptual, Language ID, Vitamin C, Syllogisms, Misconceptions, Logical Deduction, Winowhy, Novel Concepts, following Sanh et al. (2021). For English NLP tasks, in addition to 11 unseen evaluation datasets from Sanh et al. (2021), we add 3 unseen question-answering datasets from Lin et al. (2022), resulting in 7 classification (RTE (Dagan et al., 2005), CB(De Marneffe et al., 2019), ANLI R1,R2,R3 (Nie et al., 2020) WSC (Levesque et al., 2012), WiC (Pilehvar & Camacho-Collados, 2019)) and 7 multi-choice datasets (COPA (Roemmele et al., 2011), Hellaswag (Zellers et al., 2019), Storycloze (Mostafazadeh et al., 2016), PIQA (Bisk et al., 2020), ARC-Challenge (Clark et al., 2018), OpenbookQA (Mihaylov et al., 2018)). We exclude SQuAD2.0 which is included in evaluation setting of Lin et al. (2022) because it does not have label options.

## F  EVALUATION SETTING

For the result of PaLM and GPT-3 of Table 1, we use the performance reported in the paper. For the result of GPT-3 on zero-shot setting in Table 2, we use the performance reported in the paper for multi-choice tasks while we rerun the experiments using OpenAI API for classification tasks to report F1 scores. We used the prompt named 'GPT-3 style' for every dataset of Promtpsource library. For experiments of Figure 3, we randomly sample 1,000 data instances for seen task label generalization evaluation, for efficiency.

## G  CALIBRATION RESULTS

Previous work has used calibration methods to match the label distribution of the target task during inference of zero-shot setting (Zhao et al., 2021; Holtzman et al., 2021). We also analyze if calibration is effective for meta-trained LMs by applying contextual calibration on T0-3B. Because we evaluate the zero-shot task generalization performance, we use the probability of the label given an empty string for calibration. As shown in Table 3, applying calibration *hurts* the performance of meta-trained LMs.

| | |
|---|---|
| yes | no |
| true | false |
| positive | negative |
| right | wrong |
| correct | incorrect |
| agree | disagree |
| good | bad |
| guaranteed | impossible |
| always | never |
| affirmative | contradicting |
| exactly | not ever |
| undoubtedly | not at all |
| fine | disagreeable |
| good enough | cannot be |
| definitely | never |
| unquestionable | no way |
| yep | nope |
| yea | nah |
| without doubt | refused |
| willing | unwilling |

Table 4: List of 20 pairs of labels used to evaluate label generalization on binary classification datasets (RTE, WiC, IMDB, PAWS).

# H  LABEL PAIR VARIATIONS

We provide the list of variations of label pairs on Table 4 and Table 5. Table 4 shows the label pair variation of binary classification datasets (RTE, WiC, IMDB, PAWS) while Table 5 shows the label pair variation of CB, which consists of 3 label options.

| | | |
|:---:|:---:|:---:|
| yes | no | maybe |
| true | false | neither |
| positive | negative | inconclusive |
| right | wrong | perhaps |
| correct | incorrect | might be |
| agree | disagree | could be |
| good | bad | neutral |
| guaranteed | impossible | possible |
| always | never | sometimes |
| affirmative | contradicting | feasible |
| exactly | not ever | as it may be |
| undoubtedly | not at all | doubtfully |
| fine | disagreeable | conceivable |
| good enough | cannot be | can be |
| definitely | never | uncertain |
| unquestionable | no way | questionable |
| yep | nope | iffy |
| yea | nah | nn |
| without doubt | refused | controversial |
| willing | unwilling | not for sure |

Table 5: List of 20 pairs of labels used to evaluate label generalization for CB, which has 3 label options.

