# OpenReview forum: "Guess the Instruction! Flipped Learning Makes Language Models Stronger Zero-Shot Learners"
_ICLR.cc/2023/Conference — ICLR 2023 poster_

### Official Review · Reviewer_dMHf · 2022-10-25

**Confidence:** 4
**Correctness:** 3
**Technical Novelty And Significance:** 2
**Empirical Novelty And Significance:** 3
**Recommendation:** 6

**Clarity, Quality, Novelty And Reproducibility:**

Clarity: The paper is quite well written

Quality: A good amount of experiments are done to demonstrate the applicability

Novelty: Low to moderate. (low novelty is fine if the method is simple and really work well)

Reproducibility: Due to the simplicity, if should be quite easy to reproduce.

**Strength And Weaknesses:**

Pros:

The simplicity of this approach is a plus. The avoidance of label overfitting due to FLIP is also a nice advantage with additional benefits on unseen labels.


Questions:
- Any additional thoughts on why unlikelihood training is crucial for the success of FLIPPED?

Cons:

- It is not surprising that the meta training done will make 3B model outperform larger models. The statement that the model performs better compared to 175 or 540B models do not mean much and such statement should be avoided or with caveats mentioned explicitly. To compare with the larger models, it would be good to see the few-shot results without meta training. Or is it the case that FLIPPED does not work at all without meta training?
- In approach makes sense, however, I still question the universality of this approach. The free-form generation of label being one.
- Another strong baseline for tasks with unseen labels is to add label explanation in the instructions.
- Performance gain is observed for some datasets but not overwhelmingly consistent.



**Summary Of The Paper:**

The paper proposed an approach to improve zero-shot prediction results by making a language model predict the instruction (or measure likelihood) given the label and the input, rather than having the model predict the label instead. The paper demonstrates the effectiveness on multiple datasets within the BIG-Bench benchmark.

**Summary Of The Review:**

Overall, this paper proposes a simple approach consisting of the flipped inference along with the unlikelihood meta training that seems to work well. The experiments are quite extensive. The performance gain is not too consistent so while this method might work in some domain, it might be challenging to be adopted as a universal method.

---

> ### Author Response · Authors · 2022-11-11
> **Response to Reviewer dMHf**
>
> We appreciate your detailed comments about our paper.
>
> ### Response to Cons:
> > 1. The statement that the model performs better compared to 175 or 540B models do not mean much and such statement should be avoided or with caveats mentioned explicitly.
>
> We agree that while comparing FLIPPED with non-meta-trained LMs with a much larger scale, we did not mention it explicitly. We will clearly mention the differentiation of meta-trained vs non-meta-trained LMs. For comparing meta-trained LMs with much larger non-meta-trained LMs, we followed the evaluation setting of [Sanh et al (2021)](https://openreview.net/forum?id=9Vrb9D0WI4), which also compares the performance of meta-trained T0 3B to the zero-shot inference of GPT-3 175B. Moreover, although non-meta-trained LMs relatively show poor zero-shot performance, they are strong few-shot in–context learners ([Brown et al (2020)](https://arxiv.org/abs/2005.14165), [Chowdhery et al (2022)](https://arxiv.org/abs/2204.02311)). Therefore, we believe the result that FLIPPED 3B zero-shot even outperforms 3-shot inference of GPT-3 175B on BIG-bench mean accuracy is a nontrivial finding.
>
> > 2. To compare with the larger models, it would be good to see the few-shot results without meta training. Or is it the case that FLIPPED does not work at all without meta training?
>
> Without flipped learning, evaluation through calculating the likelihood of instruction given input instance and label option is not effective because the pretrained backbone LM itself has not learned the strong correspondence between the input instance and label options.
>
> > 3. In approach makes sense, however, I still question the universality of this approach. The free-form generation of label being one.
>
> We agree that FLIPPED alone has limited applicability compared to DIRECT models. Please refer to the above “Common response to reviewers” comment for more details.
>
> > 4. Another strong baseline for tasks with unseen labels is to add label explanation in the instructions.
>
> For analysis with unseen labels shown in Figure 3, the label options are included in the instruction to guide meta-trained LMs to predict one of the label options. (Ex) We use the instruction of “{{premise}} Using only the above description and what you know about the world, is "{{hypothesis}}" definitely correct? Right or Wrong?” instead of “{{premise}} Using only the above description and what you know about the world, is "{{hypothesis}}" definitely correct?”) In this experimental setting, all models except FLIPPED show low mean and high variance. Although there could be more methods for label explanation, it requires extensive editing of instructions, which can be future work.
>
> > 5. Performance gain is observed for some datasets but not overwhelmingly consistent.
>
> We agree that the result of FLIPPED 3B is not overwhelmingly consistent. However, for BIG-bench, scaling the model size (FLIPPED 11B)  leads to more consistent improvements across various datasets, outperforming T0 on 11 out of 14 datasets. For English NLP tasks, FLIPPED shows consistent improvement compared to T0 for datasets containing unseen label options (excluding RTE, WSC, WiC): 10 out of 11 datasets for both 3B and 11B scale. We think that performance on unseen labels is closer to a true task generalization setting.
>
> ### Response to Questions:
> > Any additional thoughts on why unlikelihood training is crucial for the success of FLIPPED?
>
> We conjecture that the effect of unlikelihood training is the result of avoiding shortcut learning of LLMs. For direct models, if the input instance-label correspondence is not considered, the training objective will be not optimized. Unlike direct models, FLIPPED without unlikelihood training can optimize the objective even without considering input instance-label correspondence; the loss would be also optimized if the LLM generates the instruction of the task regardless of the label option. (ex) <positive_review> negative ⇒ What is the sentiment expressed in the task?) As also shown by [Min et al (2022)](https://arxiv.org/abs/2202.12837), meta-training encourages the model to ignore complex aspects such as input instance-label correspondence, and exploit simpler aspects. Therefore, to prevent the model from exploiting simpler aspects to optimize the training objective (shortcut learning), we additionally do unlikelihood training.

---

### Official Review · Reviewer_CFao · 2022-10-25

**Confidence:** 4
**Correctness:** 3
**Technical Novelty And Significance:** 2
**Empirical Novelty And Significance:** 2
**Recommendation:** 6

**Clarity, Quality, Novelty And Reproducibility:**

The paper is mostly well-written and easy to follow. Just a few comments:
* The reason as to why Flipped Training improves label generalization is not very clear. Why would conditioning on a label avoid overfitting on it?
* What percentage of the labels in the tasks from Big-Bench and the other 14 English NLP tasks are not seen?
* The unlikelihood loss in section 3.2 could benefit from an example for more clarity.
* It would also be interesting to see how the model after Flipped Training performs on non-classification tasks? Is it going to be worse or better? Any thoughts on this?


**Strength And Weaknesses:**

### Strength
Computing the conditional probability of instruction given the concatenation of the input and label has led to strong zero-shot generalization performance. It is mentioned that this is due to the model’s better generalization to labels that it has not seen before (labels that have different surface form, but similar meaning).

### Weaknesses
As the authors mention these as well, their proposed training method is limited to tasks which have label options (vs. free-form generation). This would limit the approach in both training and inference/evaluation. It is also not always the case that one could have a task instruction that is easily separable from the input instance.


**Summary Of The Paper:**

This paper proposes a new meta-training approach for language models. For an instance of task instruction, input and label, their method, called Flipped Learning, increases the likelihood of the task instruction given the input and correct label, while decreasing the likelihood of the task instruction given the input and an incorrect label. Training a T5-3B model with this method on 20 datasets leads to performance gains on various datasets compared to larger (size and compute) models.

**Summary Of The Review:**

Overall, the approach of conditioning on the label space instead of generating it is interesting, but limited. The motivation for this method could have been discussed more. Some parts about the intuition of why the method increases label generalization are not clear.

---

> ### Author Response · Authors · 2022-11-11
> **Response to Reviewer CFao**
>
> We appreciate your overall thoughtful feedback on our paper.
>
> ### Response to Weakness:
> > 1. As the authors mention these as well, their proposed training method is limited to tasks which have label options (vs. free-form generation). This would limit the approach in both training and inference/evaluation.
>
> We have additionally trained FLIPPED+ by adding generation tasks for flipped learning, using 52 datasets of the T0++ model. For generation task training, we select an incorrect option by randomly selecting a different answer from the training set. FLIPPED+ shows improved performance compared to FLIPPED on the MMLU benchmark. Please refer to the above “Common response to reviewers” comment for more details.
>
> > 2. It is also not always the case that one could have a task instruction that is easily separable from the input instance.
>
> We agree that the underlying assumption of flipped learning is that the task instruction and input instance are easily separable. However, from preliminary experiments, we found that not separating the task instruction and input instance and putting a general instruction such as “Answer to the following question.” on the output space also works well.
>
>
> ### Response to Comments:
>
> > 1. The reason as to why Flipped Training improves label generalization is not very clear. Why would conditioning on a label avoid overfitting on it?
>
> [Min et al (2022)](https://arxiv.org/abs/2202.12837) have shown that meta-trained LMs are more likely to exploit the space that they are trained to generate on rather than the space that they are trained to condition on.  [Min et al (2022)](https://arxiv.org/abs/2202.12837) support this finding by showing that the “Channel MetaICL” model is more robust to variations on the label space than the “Direct MetaICL”. Similar to this finding, we find that FLIPPED, which is conditioned on label space, is more robust to variations in the label space than the DIRECT model.
>
> > 2. The motivation for this method could have been discussed more.
>
> It can be seen that FLIPPED is an improved version of the noisy channel model [(Min et al, 2021)](https://arxiv.org/abs/2108.04106) for a zero-shot setting. Although the channel model has the advantage that it can potentially generalize to unseen labels by avoiding label overfitting, the setup is often not well-defined for zero-shot settings. From the example of Figure 1 from our paper, the meta-trained channel LM has to generate both the input instance and instruction given only the label option “Yes”.  However, FLIPPED only generates the instruction when given input instance and label option, which is a more well-defined setup for zero-shot setting. Therefore, from this perspective, it can be seen that FLIPPED is a well-defined setup of the noisy channel model.
>
> > 3. What percentage of the labels in the tasks from Big-Bench and the other 14 English NLP tasks are not seen?
>
> For 14 English NLP tasks, 3 tasks (RTE, WSC, WiC) do not have any unseen labels (0%), 4 tasks (CB, ANLI R1, R2, R3) contain some unseen labels (33%), and the other 8 tasks contain only unseen labels (100%). For Big-bench tasks, only StrategyQA does not have any unseen labels (0%), Vitamin contains some unseen labels (33%), and the remaining 12 tasks contain only unseen labels (100%).
>
> > 4. The unlikelihood loss in section 3.2 could benefit from an example for more clarity.
>
> We thank you for your suggestion. We have added an illustration of unlikelihood loss in Figure 6 in the updated version for clarification.
>
> > 5. It would also be interesting to see how the model after Flipped Training performs on non-classification tasks? Is it going to be worse or better? Any thoughts on this?
>
> As shown in Figure 5, Flipped learning is optimized for tasks that have label options. Therefore, it is nontrivial to evaluate FLIPPED on non-classification tasks without any label options. We leave the application of FLIPPED to generation tasks as future work. Please refer to the above “Common response to reviewers” comment for more details.

---

### Official Review · Reviewer_55Bv · 2022-10-26

**Confidence:** 5
**Correctness:** 4
**Technical Novelty And Significance:** 4
**Empirical Novelty And Significance:** 4
**Recommendation:** 8

**Clarity, Quality, Novelty And Reproducibility:**

This paper is well-written, the proposed method is novel, the experiments are well-documented and should be reproducible.


**Strength And Weaknesses:**

### Strengths

1. The proposed method is novel and neat
2. The empirical results are strong demonstrated by a comprehensive set of experiments. As a new way of instruction tuning, FLIPPED could be potentially very impactful given that it outperforms much larger models with much smaller training costs.
3. The analysis is interesting and insightful, revealing that T0-like methods have difficulties generalizing to unseen label options while FLIPPED does a pretty good job.
4. The paper is well-written.

### Weaknesses

As mentioned in the Limitation section, FLIPPED is only naturally applicable to classification tasks where the label space is limited, while DIRECT methods appear to be more flexible from this aspect.


### Suggestions
When presenting the main results, I think the authors should clarify FLIPPED outperforms DIRECT on X out of X datasets. The gains are actually not very consistent in the Tables and the mean accuracy does not tell the full story.


**Summary Of The Paper:**

This paper focuses on the zero-shot task generalization setting and proposes to learn to generate the instruction conditioned on the input and label. During inference, labels can be predicted by checking which label is the most likely to generate the given instruction. The authors also incorporate unlikelihood training loss to overcome the degeneration issue where the label becomes uncorrelated with the instruction generation. Such a reversed way of generation achieves very good results on multiple unseen tasks, outperforming the T0 baseline. Notably, a 3B T5 model trained with the proposed method even outperforms T0-11B on 14 BIG-bench tasks and much larger GPT3 and PaLM in zero-shot settings. Analysis shows that the proposed method is mainly beneficial for tasks where the label options are unseen during training.

**Summary Of The Review:**

I recommend acceptance of this paper since the proposed method is novel and simple and achieves very strong results on impactful benchmarks.

---

> ### Author Response · Authors · 2022-11-11
> **Response to Reviewer 55Bv**
>
> We appreciate your helpful comments on our paper.
>
> ### Response to Weakness:
>
> > As mentioned in the limitation section, FLIPPED is only naturally applicable to classification tasks where the label space is limited, while DIRECT methods appear to be more flexible from this aspect.
>
> We agree that FLIPPED alone is only naturally applicable to classification tasks where the label space is limited. However, by adding a separate model to generate the label options, FLIPPED is also applicable to generation tasks. Please refer to the above “Common response to reviewers” comment for more details.
>
>
> ### Response to Suggestions:
> > When presenting the main results, I think the authors should clarify FLIPPED outperforms DIRECT on X out of X datasets. The gains are actually not very consistent in the Tables and the mean accuracy does not tell the full story.
>
> We thank you for your helpful suggestion. We agree that mean accuracy alone does not tell the full story. For the updated version of the paper, we have added the number of datasets that FLIPPED outperforms baselines (DIRECT, T0) for clarification.

---

### Author Response · Authors · 2022-11-11
**Common response to reviewers (Updates and Common concerns)**

We thank all three reviewers for the detailed feedback on our paper. Here are some updates on the paper as well as responses to common concerns about the paper.

### Major Updates:

We have also trained a FLIPPED (11B) model using the same training configuration as FLIPPED (3B). Our results show that FLIPPED shows scaling laws for both BIG-bench tasks and English NLP tasks, further showing that our approach is scalable. FLIPPED (11B) outperforms T0 (11B) by 8.38 mean accuracy for BIG-bench and 1.73 mean accuracy for English NLP tasks. Please refer to the additional result in Figure2, Table 1, and Table 2 results for detail.

We have also added the MMLU benchmark [(Hendrycks et al, 2021)](https://openreview.net/forum?id=d7KBjmI3GmQ) as additional evaluation datasets. Results show that FLIPPED shows consistent improvement compared to T0 for both 3B and 11B. Moreover, we have also additionally provided results for FLIPPED+, which uses more training datasets during meta-training: 52 training datasets from T0++. FLIPPED+ improves the performance of FLIPPED and outperforms T0 with the same scale with 12.12, 6.86 points for 3B and 11B model respectively. This is consistent with the findings of a concurrent work [(Chung et al, 2022)](https://arxiv.org/abs/2210.11416), which shows that scaling the number of training tasks during meta-training is effective.

MMLU Zero-shot performance:

| Model |   T0  | FLIPPED |  FLIPPED+ |   T0  | FLIPPED | FLIPPED+ |
|:-----:|:-----:|:-------:|:---------:|:-----:|:-------:|:-------:|
|  Size |   3B  |    3B   |     3B    |  11B  |   11B   |  11B  |
|  ACC  | 35.11 |  38.05  | **47.23** | 43.04 |  43.95  | **49.90**|

 ### Response to Common Concerns:

All three reviewers commonly pointed out that FLIPPED is only naturally applicable to tasks with label options, as mentioned in the limitation section of our paper. Although it is nontrivial to apply FLIPPED model for generation tasks, it can still be applied by adding a decoder-only LM beforehand to obtain possible options for the prompted instance. Previous works adopt this architecture for question-answering [(Lewis and Fan, 2019)](https://openreview.net/forum?id=Bkx0RjA9tX) and reasoning tasks [(Cobbe et al, 2021)](https://arxiv.org/abs/2110.14168). However, we leave using the FLIPPED model as a verifier for generation tasks as future work.

---

### Author Response · Authors · 2022-12-12
**Additional Results**

Sorry for an additional ping, but we have added a new result of FLIPPED+ (11B). Please refer to “Common response to reviewers” for more detail.

---

### Decision · Program_Chairs · 2023-01-20

**Decision:**

Accept: poster

**Justification For Why Not Higher Score:**

NA

**Justification For Why Not Lower Score:**

Good paper

**Metareview: Summary, Strengths And Weaknesses:**

The proposed method FLIPPED is simple and sound. It shows strong empirical results. The authors also addressed the reviewers' comments and updated the paper with new results. I am in favour of accepting the paper.

**Note From Pc:**

if the above contains the word "oral" or "spotlight" please see: "oral" presentation means -> notable-top-5% and "spotlight" means -> notable-top-25%. As stated in our emails, we are disassociating presentation type from AC recommendations

**Summary Of Ac-Reviewer Meeting:**

N/A